# 25-Hydroxycholesterol Induces Intrinsic Apoptosis via Mitochondrial Pathway in BE(2)-C Human Neuroblastoma Cells

**DOI:** 10.3390/ijms26168012

**Published:** 2025-08-19

**Authors:** Jaesung Kim, Koanhoi Kim, Dongha Park, Seong-Kug Eo, Bo-Ae Lee, Yonghae Son

**Affiliations:** 1Department of Pharmacology, School of Medicine, Pusan National University, Yangsan 50612, Republic of Korea; rlawotjd1029@naver.com (J.K.); koanhoi@pusan.ac.kr (K.K.); propertopic@pusan.ac.kr (D.P.); 2College of Veterinary Medicine and Bio-Safety Research Institute, Jeonbuk National University, Iksan 54896, Republic of Korea; vetvirus@chonbuk.ac.kr; 3Department of Physical Education, Dongguk University, Seoul 04620, Republic of Korea

**Keywords:** 25-hydroxycholesterol, neuro-inflammation, neuronal cell death, oxysterol, mitochondria

## Abstract

An oxysterol, 25-Hydroxycholesterol (25OHChol), is produced through cholesterol oxidation and is involved in various cellular processes, including apoptosis. However, the precise mechanisms underlying 25OHChol-induced apoptosis in neuroblastoma cells remain unclear. The aim of this study was to elucidate the detailed molecular mechanisms by which 25OHChol induces apoptosis in human neuroblastoma cells. This study explores the apoptotic effects of 25OHChol and the associated signaling pathways in BE(2)-C cells, a widely used human neuroblastoma cell model for neuronal differentiation and cancer research. To evaluate the cytotoxicity of 25OHChol, cell viability was assessed using the CCK-8 assay, which demonstrated a concentration-dependent decline, indicating a potential induction of cell death. Morphological changes characteristic of apoptosis, such as nuclear condensation and fragmentation, were confirmed via DAPI staining. Additionally, Annexin V/PI flow cytometry analysis revealed an increase in late apoptotic cell populations, further corroborating apoptosis induction. To investigate the molecular mechanisms, we analyzed the expression of Bcl-2 family proteins via Western blotting. The results showed an elevated Bax/Bcl-2 ratio, suggesting activation of the intrinsic mitochondrial apoptotic pathway. This was further supported by a reduction in mitochondrial membrane potential (MMP), as measured by flow cytometry. Increased caspase-9 and caspase-3/7 activity provided additional evidence for caspase-mediated apoptosis. Moreover, treatment with the pan-caspase inhibitor Z-VAD-FMK led to a dose-dependent increase in cell viability, confirming the essential role of caspases in 25OHChol-induced apoptosis. In conclusion, this study demonstrates that 25OHChol triggers apoptosis in BE(2)-C neuroblastoma cells through activation of the intrinsic mitochondrial apoptotic pathway. These findings provide new insights into the cytotoxic effects of 25OHChol and its potential role in neuroblastoma cell death.

## 1. Introduction

Neuroblastoma is the most common extracranial solid tumor in children, accounting for approximately 8–10% of pediatric malignancies and 15% of childhood cancer-related deaths [1]. This tumor arises from undifferentiated neural crest cells within the peripheral sympathetic nervous system, adrenal medulla, or paraspinal ganglia [2]. Its incidence is estimated at 1 in 10,000 live births, with roughly 500 new cases diagnosed annually in the United States. Neuroblastoma primarily affects young children, with a median age at diagnosis of 22 months, and approximately 90% of cases are diagnosed before the age of 10 [3,4].

Early symptoms are often nonspecific—fatigue, loss of appetite, fever, and joint pain—contributing to delayed diagnosis. By the time of diagnosis, nearly 70% of high-risk patients present with metastatic disease, which significantly limits treatment options and leads to poor prognoses [5]. Although intensive multimodal therapies—including chemotherapy, radiation, surgery, bone marrow transplantation, and biologic treatments—have improved survival rates in some cases [6], the overall survival rate for high-risk neuroblastoma remains low. Frequent relapses and the lack of a curative therapy for advanced-stage disease further complicate treatment [7]. Additionally, these aggressive interventions are associated with severe short- and long-term toxicities, including secondary malignancies [8]. Therefore, there is an urgent need to develop more effective and less toxic therapeutic strategies for high-risk neuroblastoma.

Cell death is generally classified into non-programmed and programmed cell death (PCD) [9]. Non-programmed cell death, predominantly necrosis, occurs passively in response to external damage, leading to cell membrane rupture and the release of intracellular contents, which often trigger inflammation [10]. In contrast, PCD is a tightly regulated process that allows cells to self-destruct in a controlled manner, playing a critical role in tissue homeostasis [11]. Among the various forms of PCD, apoptosis is particularly important as it facilitates cellular clearance without provoking inflammation [12]. Dysregulation of apoptosis is implicated in various diseases, including cancer and neurodegenerative disorders [13,14].

Apoptosis occurs through two major pathways: the intrinsic (mitochondrial) pathway and the extrinsic (death receptor-mediated) pathway [15]. The intrinsic pathway is regulated by Bcl-2 family proteins, which determine mitochondrial membrane permeability. Anti-apoptotic proteins, such as Bcl-2, inhibit apoptosis, whereas pro-apoptotic members, such as Bax and Bak, promote cell death [16,17]. Upon activation, Bax and Bak increase mitochondrial permeability, leading to cytochrome c release into the cytoplasm [18]. Cytochrome c then binds with apoptotic protease-activating factor-1 (Apaf-1) to form the apoptosome, which activates caspase-9 and initiates a caspase cascade, culminating in caspase-3 activation [19,20].

Cholesterol (Chol) is an essential structural component of mammalian cell membranes, regulating membrane fluidity and permeability, facilitating membrane trafficking and signal transduction, and serving as a precursor for steroid hormones [21]. Oxysterols, which are oxidized derivatives of Chol, are generated through enzymatic or non-enzymatic oxidation [22]. Enzymatic oxidation is mediated by specific hydroxylases, whereas non-enzymatic oxidation primarily results from reactive oxygen species [23]. Oxysterols are categorized as either primary, which are directly derived from Chol (e.g., 24s-hydroxycholesterol [24sOHChol], 25-hydroxycholesterol [25OHChol], and 27-hydroxycholesterol [27OHChol]), or secondary, which are generated from primary oxysterols [24].

Recent research has highlighted the biological significance of side-chain oxysterols, including 24sOHChol, 25OHChol, and 27OHChol, due to their distinct cellular effects [25,26,27]. For example, 24sOHChol exhibits cytotoxic effects on neuronal cells and has been implicated in neurodegenerative diseases [28]. It also induces apoptosis and necroptosis in SH-SY5Y neuroblastoma cells and lymphoma cells [29,30]. Similarly, 27OHChol promotes apoptosis in hematopoietic progenitor cells and myeloid cell lines [31]. Several studies have also demonstrated that 25OHChol induces apoptosis in various cancer cell types, including monocytic cells, pheochromocytoma cells, and head and neck squamous carcinoma cells, and it has been shown to exert tumor-suppressive effects in leukemic cells [32,33,34,35]. These oxidized Chol derivatives exhibit significant cytotoxicity toward both normal and malignant cells, with effects depending on cell type and oxysterol concentration [36,37]. Due to their pro-apoptotic and anti-tumor properties, oxysterols have been proposed as potential therapeutic agents for cancer treatment [38].

Despite growing evidence for their anticancer properties in various tumor types, the biological effects of oxysterols on neuroblastoma cells have not been systematically examined. In this context, the present study was designed to address this knowledge gap by focusing on the interaction between oxysterols and neuroblastoma cell biology, with particular emphasis on potential apoptotic pathways.

## 2. Results

### 2.1. Viability of BE(2)-C Cells Following Oxysterol Treatment

To evaluate the impact of 25OHChol on the viability of BE(2)-C cells, the cells were treated with Chol or oxysterols at concentrations of 0.5, 1, and 2 µg/mL for 24, 48, and 72 h. Cell viability was subsequently assessed using the CCK-8 assay. As shown in Figure 1A, only treatment with 25OHChol led to a significant reduction in viability, decreasing to 61.7%, whereas other treatments had no noticeable effect. Furthermore, the results presented in Figure 1B demonstrate that 25OHChol induced a concentration- and time-dependent decline in cell viability. Specifically, after 48 h of treatment with 25OHChol at concentrations of 0.5, 1, and 2 µg/mL, cell viability decreased to 92.1%, 58.1%, and 40.7%, respectively. Additionally, treatment with 1 µg/mL of 25OHChol for 24, 48, and 72 h resulted in viability levels of 87.1%, 50.6%, and 38.2%, respectively. These findings indicate that 25OHChol exhibits cytotoxic effects on BE(2)-C cells.

### 2.2. Induction of Apoptosis in BE(2)-C Cells by 25OHChol

Morphological changes in BE(2)-C cells were examined following treatment with Chol or oxysterols (1 µg/mL) for 48 h. As shown in Figure 2A, optical microscopy revealed that treatment with 25OHChol resulted in significant cell shrinkage and loss of cell–cell adhesion. Furthermore, DAPI staining (Figure 2B) indicated chromatin condensation and nuclear fragmentation, both hallmarks of apoptosis (white arrows).

To further characterize the stages of cell death in BE(2)-C cells, Annexin V/PI staining was performed. Annexin V binding to phosphatidylserine on the outer cell membrane indicated early apoptosis, while propidium iodide (PI) staining identified cells with compromised membranes, marking late apoptosis or necrosis. BE(2)-C cells were treated with Chol or oxysterols (1 µg/mL) for 48 h and subsequently analyzed by flow cytometry (Figure 3). Treatment with 25OHChol led to a significant increase in the apoptotic rate compared to other treatment groups. In the 25OHChol-treated group, the combined rate of early and late apoptosis reached 79.17%, whereas the control, Chol, 24sOHChol, and 27OHChol groups exhibited relatively low apoptosis rates of 6.82%, 6.74%, and 9.86%, respectively. These findings suggest that 25OHChol plays a critical role in apoptosis induction in BE(2)-C cells.

### 2.3. Role of 25OHChol in Apoptosis Through Mitochondrial Apoptotic Regulators in BE(2)-C Cells

Bax and Bcl-2 are key regulators of apoptosis, with Bax acting as a pro-apoptotic protein that promotes cell death by increasing mitochondrial membrane permeability, while Bcl-2 functions as an anti-apoptotic protein that counteracts this effect to support cell survival. To assess the expression levels of these proteins, Western blot analysis was performed after treating cells with 25OHChol for 0, 24, and 48 h, followed by quantification of protein band intensities (Figure 4A). The Bax/Bcl-2 ratio, represented graphically, exhibited a time-dependent increase, with values of 0.51, 0.94, and 1.69 at 0, 24, and 48 h, respectively (Figure 4B).

To determine whether apoptosis occurs via the mitochondria-dependent pathway, a mitochondrial membrane potential (MMP) assay was conducted using JC-1 dye. MMP loss was measured by flow cytometry following treatment with Chol or oxysterols (Figure 5A). The mean percentages of MMP loss were 17.8% in the control group, 17% with Chol, 21.5% with 24sOHChol, and 23% with 27OHChol. However, treatment with 25OHChol resulted in a significant increase in MMP loss to 33.7% (Figure 5B). These findings suggest that 25OHChol induces apoptosis in BE(2)-C cells through a mitochondria-dependent pathway, as evidenced by the elevated Bax/Bcl-2 ratio and the substantial loss of mitochondrial membrane potential.

### 2.4. Caspase-Dependent Apoptotic Effects Induced by 25OHChol in BE(2)-C Cells

Caspase-9 is an initiator caspase that plays a pivotal role in the intrinsic apoptotic pathway by activating downstream effector caspases, including caspase-3. Caspase-3, a key effector caspase, mediates the execution phase of apoptosis, leading to the cleavage of cellular components and eventual cell death. To evaluate the effects of 25OHChol on caspase activity, BE(2)-C cells were treated with Chol or oxysterols (1 µg/mL), followed by caspase activity assessment via flow cytometry and Western blot analysis. Flow cytometry results revealed that caspase-9 activity levels in the control, Chol, 24sOHChol, 25OHChol, and 27OHChol groups were 3.75%, 3.14%, 5.33%, 24.58%, and 8.73%, respectively (Figure 6A). Similarly, caspase-3/7 activity levels in these groups were 5.16%, 7.99%, 8.19%, 31.53%, and 10.83%, respectively (Figure 6B). To further investigate the time-dependent activation of caspase-3, cleaved caspase-3/pro-caspase-3 levels were analyzed by Western blot following treatment with 25OHChol for 0, 24, and 48 h. Quantification of band intensity (Figure 6C) showed a progressive increase in the cleaved caspase-3/pro-caspase-3 ratio, reaching 0.23, 0.58, and 1.13, respectively. To determine whether caspase activation contributed to 25OHChol-induced cytotoxicity, the pan-caspase inhibitor Z-VAD-FMK was used, and its effects on cell viability were assessed using the CCK-8 assay (Figure 6D). Treatment with 25OHChol alone reduced cell viability to 58.75% compared to the control. However, co-treatment with 25OHChol and Z-VAD-FMK at concentrations of 25 µM, 50 µM, and 100 µM significantly restored viability to 67.8%, 71.6%, and 78.1%, respectively. These findings demonstrate that 25OHChol induces apoptosis in BE(2)-C cells via the caspase-dependent intrinsic pathway, and this effect is attenuated in a dose-dependent manner by the pan-caspase inhibitor Z-VAD-FMK.

## 3. Discussion

In this study, we demonstrate that 25OHChol significantly reduces the viability of BE(2)-C neuroblastoma cells by inducing apoptosis through the intrinsic mitochondrial pathway. Neuroblastoma is one of the most common pediatric cancers, with a poor overall survival rate, highlighting the urgent need for novel and effective therapeutic approaches [39]. Our findings suggest that 25OHChol may serve as a promising strategy for neuroblastoma treatment by selectively inducing apoptosis in BE(2)-C cells. Consistent with prior research on oxysterol-induced apoptosis, our results confirm that 25OHChol decreases cell viability in a dose- and time-dependent manner. Unlike cholesterol and other oxysterols, 25OHChol induces extensive cell death, underscoring its potential as a selective cytotoxic agent. These findings align with previous studies on the pro-apoptotic effects of 25OHChol in other cancer models, including reports demonstrating mitochondrial disruption and caspase activation in neuronal or tumor-derived cells [34], suggesting a broader therapeutic role for oxysterols in oncology. Importantly, this pro-apoptotic effect may be linked to the compound’s ability to disrupt redox homeostasis in cancer cells. Reactive oxygen species (ROS) are known to play dual roles in tumor biology, promoting either survival or cell death depending on their intracellular levels. As highlighted by a report, a moderate increase in ROS can support tumorigenesis by activating pro-survival signaling pathways, while excessive ROS accumulation leads to oxidative damage and apoptosis [40]. In this context, 25OHChol may act by pushing ROS beyond the apoptotic threshold, thereby triggering mitochondrial dysfunction and caspase activation in BE(2)-C cells. This perspective supports the concept of using pro-oxidant therapies to selectively target cancer cells with heightened oxidative stress vulnerability.

Apoptosis was verified using multiple analytical methods. DAPI staining revealed nuclear condensation and fragmentation, both hallmarks of apoptosis [12]. Annexin V/PI staining further confirmed these apoptotic changes, as phosphatidylserine externalization indicated early apoptosis, while membrane compromise marked later apoptotic stages [41]. Flow cytometric analysis of Annexin V/PI-stained cells showed a significant increase in apoptotic populations following 25OHChol treatment, reinforcing its strong pro-apoptotic effect on neuroblastoma cells. Mechanistically, the apoptotic effect of 25OHChol was linked to mitochondrial apoptotic regulators. Western blot analysis revealed a time-dependent increase in the Bax/Bcl-2 ratio, indicating a shift toward a pro-apoptotic state in mitochondria. Bax and Bcl-2 play opposing roles in apoptosis regulation; Bax enhances mitochondrial membrane permeability to promote apoptosis, whereas Bcl-2 acts as an anti-apoptotic protein that prevents cell death [42,43]. The increase in the Bax/Bcl-2 ratio, coupled with a marked loss of mitochondrial membrane potential as shown by JC-1 staining, suggests that 25OHChol disrupts mitochondrial integrity, a critical step in apoptosis activation. In addition to mitochondrial disruption, 25OHChol activated the caspase cascade. Flow cytometric analysis revealed increased caspase-9 and caspase-3/7 activity following 25OHChol treatment, confirming activation of the intrinsic apoptotic pathway. As an initiator caspase, caspase-9 activates caspase-3, the main executioner responsible for dismantling cellular structures [44,45]. Western blot analysis further demonstrated a progressive increase in cleaved caspase-3 levels, and the partial restoration of cell viability upon treatment with the pan-caspase inhibitor Z-VAD-FMK confirmed that 25OHChol induces apoptosis through a caspase-dependent, mitochondria-mediated pathway.

This study highlights the potential of 25OHChol as an apoptosis-inducing agent for neuroblastoma via the intrinsic mitochondrial pathway. Our findings demonstrate that 25OHChol reduces BE(2)-C cell viability by activating mitochondrial regulators and the caspase cascade. However, further research is needed to determine whether the extrinsic apoptotic pathway also contributes to its cytotoxic effects. Investigating the involvement of death receptor-mediated signaling in 25OHChol-induced apoptosis could provide a more comprehensive understanding of its mechanisms and expand its therapeutic potential.

Taken together, our study provides strong evidence that 25OHChol induces apoptosis in BE(2)-C neuroblastoma cells through the intrinsic mitochondrial pathway, highlighting its potential as a novel therapeutic agent. By modulating key apoptotic regulators, including Bax, Bcl-2, and caspases, 25OHChol effectively disrupts mitochondrial integrity, leading to cell death. While our findings establish the mitochondrial pathway as a primary mechanism, future studies should explore whether 25OHChol also engages extrinsic apoptotic signaling, which may further enhance its therapeutic efficacy. Additionally, in vivo studies are necessary to validate its anti-tumor effects and assess potential off-target toxicity. In conclusion, our study underscores the promise of 25OHChol as a selective and potent pro-apoptotic agent in neuroblastoma, warranting further investigation into its clinical applicability.

## 4. Materials and Methods

### 4.1. Reagents

Chol and 24sOHChol were purchased from Santa Cruz Biotechnology, Inc. (Santa Cruz, CA, USA). 25OHChol and 27OHChol were purchased from Sigma-Aldrich (St. Louis, MO, USA) and dissolved in ethanol. Z-VAD-FMK was purchased from Cell Signaling Technology, Inc. (Danvers, MA, USA) and dissolved in dimethyl sulfoxide (DMSO; Sigma-Aldrich, St. Louis, MO, USA).

### 4.2. Cell Culture and Treatment

BE(2)-C cell line was purchased from the American Type Culture Collection (ATCC; Manassas, VA, USA) and cultured in Dulbecco’s Modified Eagle’s Medium (DMEM) supplemented with 10% fetal bovine serum (FBS) and 1% penicillin-streptomycin. The cells were maintained at 37 °C in a humidified incubator with 5% CO_2_. The medium was changed every 2 to 3 days, and cells were detached using Trypsin-EDTA (Thermo Fisher Scientific, Waltham, MA, USA). Cells (1 × 10^5^ cells/mL) were serum-starved overnight in 1% FBS/DMEM, and then treated individually with Chol, 24sOHChol, 25OHChol, and 27OHChol (1 μg/mL).

### 4.3. Cell Viability Assay

Cell viability was measured using the Cell Counting Kit-8 (CCK-8; CK04-11; Dojindo, Kumamoto, Japan). BE(2)-C cells (1 × 10^4^ cells/100 μL) were seeded into each well of a 96-well plate and incubated at 37 °C in a 5% CO_2_ environment. After treatment and incubation for the required time, the tetrazolium salt, WST-8 reagent, was added to assess the cell viability, and the cells were incubated for 2 h at 37 °C in the dark. Absorbance was then measured at 450 nm.

### 4.4. Nuclear Staining

BE(2)-C cells were seeded onto coverslips placed in each well of a 6-well plate that had been coated with 0.2% gelatin. After serum starvation overnight, cells were treated with Chol or oxysterols (1 µg/mL) for 48 h. Following treatment, the medium was removed, and the cells were washed with PBS. Cells were then stained with 6-diamidino-2-phenylindole dihydrochloride (DAPI; Sigma-Aldrich, St. Louis, MO, USA) (1 µg/mL) for 15 min at room temperature. After another PBS wash, cells were fixed with 1% paraformaldehyde (PFA) for 20 min at room temperature. After a final PBS wash, mounting medium (Vector Laboratories, Burlingame, CA, USA) was dispensed onto a slide glass, the coverslip was attached, and it was secured with nail polish. Nuclear staining was observed using a confocal microscope (Olympus FV1000; Olympus, Tokyo, Japan) at 400× magnification.

### 4.5. Assay of Annexin V/PI

The Annexin V/PI assay was conducted using the Annexin V/PI apoptosis detection kit (HY-K1073; MedChemExpress, Monmouth Junction, NJ, USA). BE(2)-C cells were serum-starved overnight and then treated with Chol or oxysterols for 48 h. After treatment, the cells were trypsinized to detach, centrifuged to remove the supernatant, washed with PBS, and resuspended in 1× binding buffer (100 µL). Afterward, 5 µL each of Annexin V solution and PI solution were added. The cells were incubated in the dark at room temperature for 20 min before being analyzed by flow cytometry (CytoFLEX, Beckman Coulter, Brea, CA, USA).

### 4.6. Assessment of Mitochondrial Membrane Potential (MMP, ΔΨm)

Mitochondrial membrane potential was measured using the reagent 5,5′,6,6′-tetrachloro-1,1′,3,3′-tetraethylbenzimidazolylcarbocyanine iodide (JC-1; T3168; Life Technologies Corporation, Carlsbad, CA, USA). BE(2)-C cells (1 × 10^5^ cells/mL) were serum-starved overnight and then treated with Chol or oxysterols (1 μg/mL) for 48 h. JC-1 (5 µg/mL) was added, and incubated at 37 °C for 10 min. Then, the cells were trypsinized, centrifuged, and washed with PBS and analyzed by flow cytometry.

### 4.7. Assay of Caspase Activity

The activity of caspase-9 was measured using the Caspase-9 (Active) FITC Staining Kit (ab65615; Abcam, Cambridge, UK). BE(2)-C cells were serum-starved overnight and subsequently treated with Chol or oxysterols (1 μg/mL) for 48 h. Cells were then incubated with FITC-LEHD-FMK, a fluorescent marker for caspase-9 activity, for 1 h at 37 °C in a 5% CO_2_ atmosphere. After incubation, cells were washed and resuspended in assay buffer, then analyzed by flow cytometry. Caspase-3/7 activity was assessed using the Caspase-3/7 Activity Apoptosis Assay Kit (22796; AAT Bioquest, Sunnyvale, CA, USA). BE(2)-C cells were serum-starved overnight and treated with Chol or oxysterols (1 μg/mL) for 48 h. The cells were then incubated with TF2-DEVD-FMK as a fluorogenic marker for caspase-3/7 activity for 3 h at 37 °C in a 5% CO_2_ atmosphere. After incubation, cells were washed and resuspended in assay buffer, then analyzed by flow cytometry.

### 4.8. Western Blot Analysis

After BE(2)-C cells were serum-starved overnight, the cells were treated with 25OHChol (1 µg/mL) at various times. Protein was extracted using PRO-PREP^TM^ (iNtRON Biotechnology, Seoul, Republic of Korea). Protein was quantified using the bicinchoninic acid method (BCA, Pierce, Rockford, IL, USA), mixed with 6× loading dye, and heated at 95 °C for 1 min. 20 µg of protein in 20 µL were loaded into each well and electrophoresed at 50 V for 1 h, then at 90 V for 2 h on a 12.5% gel. The proteins were transferred to a nitrocellulose membrane (NC membrane; BIO-RAD, Hercules, CA, USA) and blocked with 5% skim milk for 1 h. Membranes were incubated overnight at 4 °C with primary antibodies against β-actin (1:3000; sc-47778; Santa Cruz Biotechnology, CA, USA), Bax/Bcl-2 (1:1000; 610982/610538; Transduction Laboratories, Lexington, KY, USA), and pro-caspase-3/cleaved capase-3 (1:1000; 9668S/9661S; Cell Signaling Technology, Inc., Danvers, MA, USA). The membranes were then washed three times for 10 min each with Tris-buffered saline containing Tween 20 (TBS-T), followed by incubation with HRP-conjugated secondary antibodies (1:5000; sc-516102; Santa Cruz Biotechnology) in TBS-T at room temperature for 1 h. After three more 10 min washes with TBS-T, the HRP substrate was added and the bands were detected using chemiluminescence with an Amersham imager (Amersham Pharmacia Biotech, Piscataway, NJ, USA).

### 4.9. Statistical Analysis

Statistical analysis was performed using one-way ANOVA, followed by Tukey’s multiple comparison test. All data were analyzed using PRISM (version 5.0; GraphPad Software Inc., San Diego, CA, USA). *p* ≤ 0.05 was considered statistically significant.

## Figures and Tables

**Figure 1 ijms-26-08012-f001:**
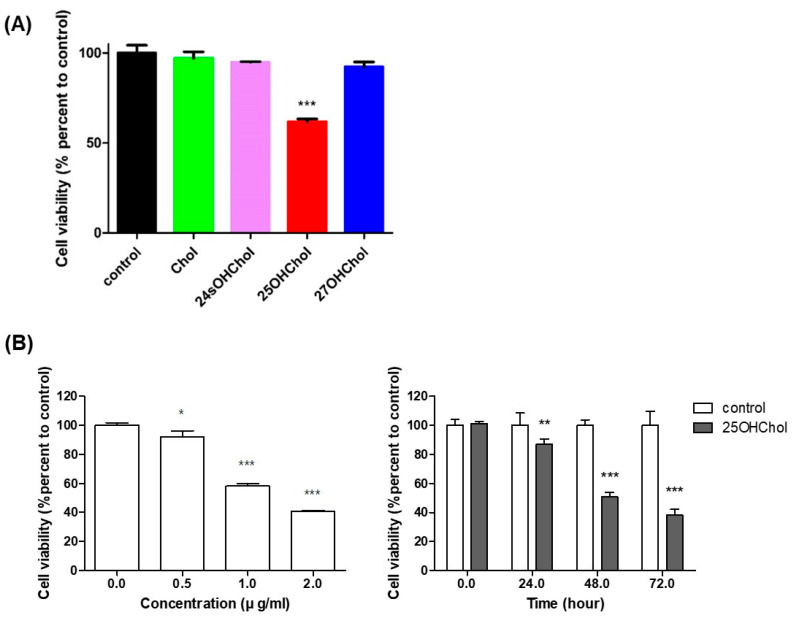
Effects of 25OHChol on viability of BE(2)-C cells. (**A**) BE(2)-C cells were treated with 1 μg/mL of Chol or oxysterols for 48 h. The cells were incubated at indicated concentrations of 25OHChol for 48 h, and at 1 μg/mL of the oxysterol for indicated times (**B**). The cell viability was measured using the CCK-8 assay, and data are presented as the mean ± SD, with three replicates per condition (n = 3). * *p* < 0.05, ** *p* < 0.01, and *** *p* < 0.001 compared to the control group.

**Figure 2 ijms-26-08012-f002:**
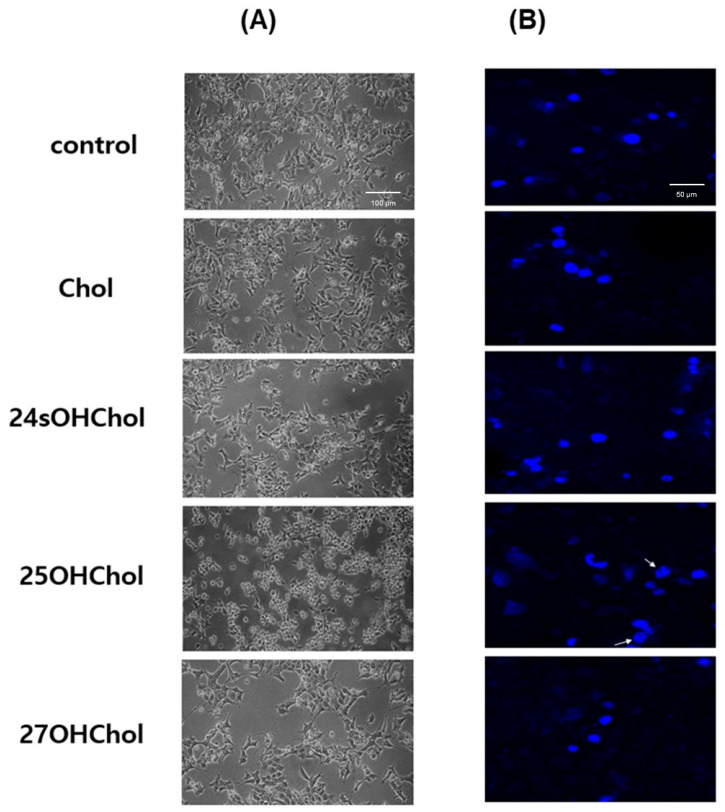
Effects of 25OHChol on morphological changes in BE(2)-C cells. BE(2)-C cells were serum-starved and then incubated for 48 h with Chol or oxysterols (1 µg/mL). (**A**) Morphological changes in BE(2)-C cells were assessed using a light microscope at 200× magnification. (**B**) Nuclear morphological changes were evaluated by staining the cells with DAPI and observing them under a confocal microscope at 400× magnification.

**Figure 3 ijms-26-08012-f003:**
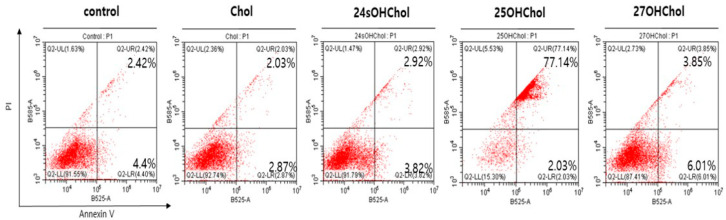
Effects of 25OHChol on apoptosis of BE(2)-C cells. BE(2)-C cells were serum-starved and then incubated for 48 h with Chol or oxysterols (1 µg/mL). Unstained and single-stained samples with Annexin V and PI were prepared to establish compensation. The cells were then double-stained with Chol or oxysterols and analyzed by flow cytometry.

**Figure 4 ijms-26-08012-f004:**
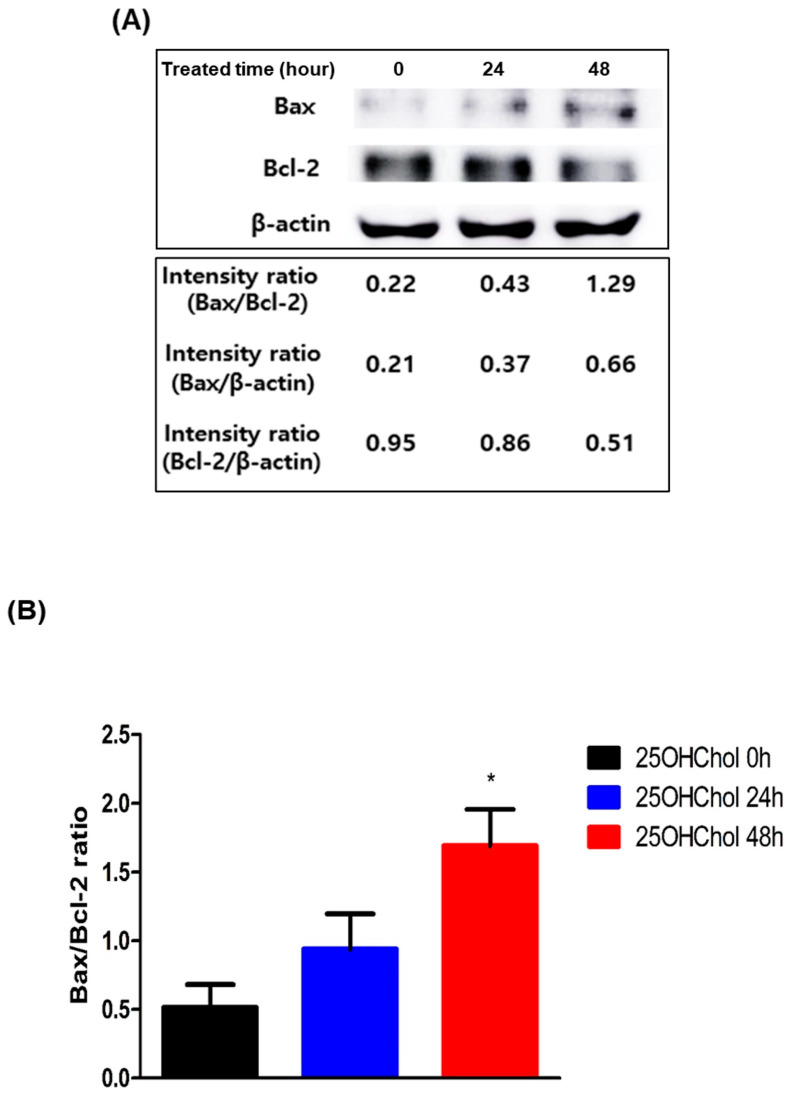
Expression of Bax and Bcl-2 proteins after treatment with 25OHChol in BE(2)-C cells. BE(2)-C cells were serum-starved and then incubated with 25OHChol (1 µg/mL) for 0, 24, and 48 h. (**A**) Cells were lysed, and total protein was analyzed via Western blot to detect Bax and Bcl-2. The band intensity ratio of Bax and Bcl-2 was calculated and presented using ImageJ software (ver.1.54p; National Institutes of Health, Bethesda, MD, USA). (**B**) The mean values of Bax/Bcl-2 ratio displayed as a graph and data are presented as the mean ± SD, with three replicates per condition (n = 3). * *p* < 0.05 compared to the control group.

**Figure 5 ijms-26-08012-f005:**
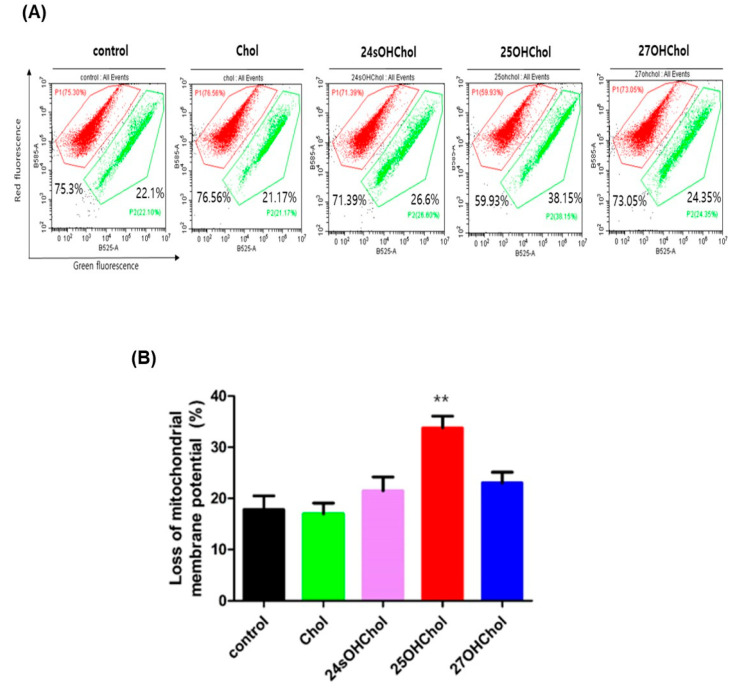
Effects of mitochondrial membrane potential (MMP) after treatment with 25OHChol in BE(2)-C cells. BE(2)-C cells were treated with Chol or oxysterols at a concentration of 1 µg/mL for 48 h. Following treatment, the cells were trypsinized and stained with JC-1 dye at 37 °C for 20 min. Mitochondrial membrane potential was then assessed using flow cytometry. (**A**) shows data from individual independent experiments, while (**B**) represents the mean values of cells with a loss of mitochondrial membrane potential. Data are presented as the mean ± SD from three replicates per condition (n = 3). ** *p* < 0.01 compared to the control group.

**Figure 6 ijms-26-08012-f006:**
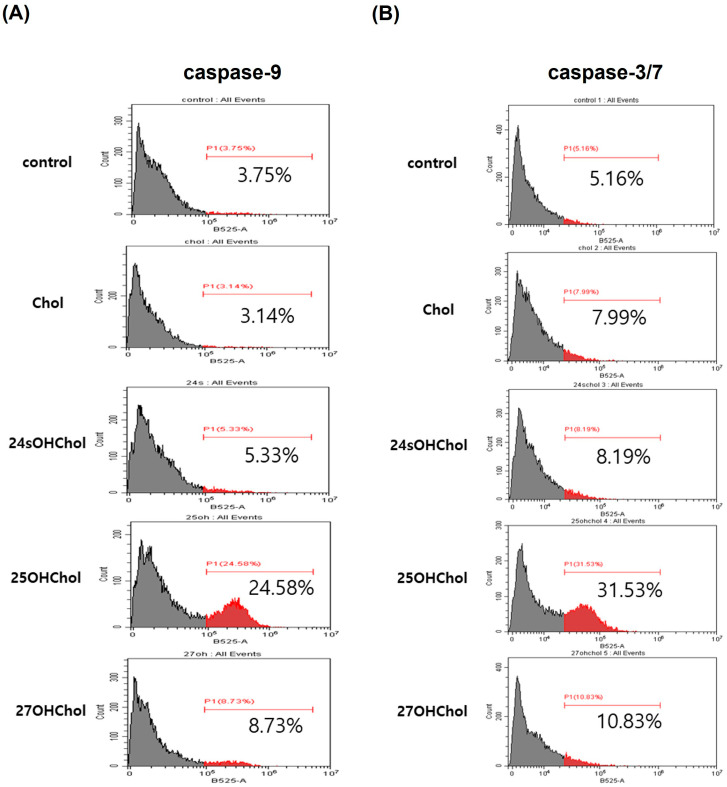
Effects of 25OHChol on caspase activity in BE(2)-C cells. BE(2)-C cells were treated with Chol or oxysterols at a concentration of 1 µg/mL for 48 h, after which caspase activity was evaluated using flow cytometry and Western blotting. (**A**) To measure caspase-9 activity, cells were incubated with FITC-LEHD-FMK at 37 °C with 5% CO_2_ for 1 h, followed by flow cytometry analysis. (**B**) For assessing caspase-3/7 activity, cells were incubated with TF2-DEVD-FMK under the same conditions for 3 h, and activity was then measured by flow cytometry. (**C**) In a separate experiment, BE(2)-C cells were treated with 25OHChol at 1 µg/mL, and cell lysates were harvested at 0, 24, and 48 h after treatment. Total protein was analyzed via Western blot to detect caspase-3. The band intensity ratio of cleaved caspase-3/pro-caspase-3 was calculated and the mean values presented as a graph, with data shown as the mean ± SD from three replicates per condition (n = 3). Statistical significance was determined at *** *p* < 0.001 compared to the control group. (**D**) BE(2)-C cells were exposed to 1 μg/mL 25OHChol and Z-VAD-FMK at specified concentrations for 48 h. Cell viability was determined using the CCK-8 assay, with data presented as the mean ± SD from three replicates per condition (n = 3). *** *p* < 0.001 compared to the control group and # *p* < 0.5, ## *p* < 0.1, ### *p* < 0.01 compared to the 25OHChol group.

## Data Availability

All data generated or analyzed during this study are included in this published article and its Appendix A.

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
