# Peer review of "25-Hydroxycholesterol Induces Intrinsic Apoptosis via Mitochondrial Pathway in BE(2)-C Human Neuroblastoma Cells"

_ijms, 2025, doi:10.3390/ijms26168012_

Round 1

Reviewer 1 Report

Comments and Suggestions for Authors

Overall, the manuscript is well written. It provides a detailed explanation of the mitochondrial assessment of possible programmed cell death in human endoblastoma cells. However, I feel that it lacks references to additional studies, such as that by Rao et al., 1999 (https://doi.org/10.1111/j.1749-6632.1999.tb07860.x), which discuss the development of apoptosis in human endoblastoma cells through the action of 24-hydroxy-cholesterol. Furthermore, the manuscript states that DAPI mounting stain was used to determine chromatin condensation and DNA fragmentation. Is this possible? Wouldn't more specific kits be required for this, such as the Hoechst kit?

Author Response

< Answer for Reviewer 1 >

Overall, the manuscript is well written. It provides a detailed explanation of the mitochondrial assessment of possible programmed cell death in human endoblastoma cells. However, I feel that it lacks references to additional studies, such as that by Rao et al., 1999 (https://doi.org/10.1111/j.1749-6632.1999.tb07860.x), which discuss the development of apoptosis in human endoblastoma cells through the action of 24-hydroxy-cholesterol. Furthermore, the manuscript states that DAPI mounting stain was used to determine chromatin condensation and DNA fragmentation. Is this possible? Wouldn't more specific kits be required for this, such as the Hoechst kit?

A: Thank you for your exceptionally thoughtful and invaluable advice. I sincerely apologize for having overlooked an important paper. Upon review, I found that the paper you mentioned had been omitted from the Introduction. I have now confirmed this, added the reference, and highlighted the revised section in blue.

You also mentioned the comparison between DAPI and Hoechst staining. The primary difference between the two is that Hoechst stains can efficiently penetrate and label live cells, whereas DAPI is less permeable to live cells and may not stain them effectively. In our case, however, the staining was performed after cell fixation, which enables DAPI to adequately label the nuclei. Nevertheless, I fully acknowledge the importance of your observation and will take extra care in future experiments.

Once again, I deeply appreciate your thoughtful and high-level feedback.

Reviewer 2 Report

Comments and Suggestions for Authors

Review of the manuscript: “25-Hydroxycholesterol Induces Intrinsic Apoptosis via Mitochondrial Pathway in BE(2)-C Human Neuroblastoma Cells”. The submitted manuscript presents a carefully designed experimental study evaluating the cytotoxic effects of 25-hydroxycholesterol (25OHChol) on BE(2)-C neuroblastoma cells. The methodologies used—CCK-8, DAPI, Annexin V/PI, JC-1 staining, caspase activity assays, and Western blotting—are appropriate and convincingly support the activation of the intrinsic apoptotic pathway. The article is scientifically sound, well-documented, and potentially valuable to the fields of neuro-oncology and oxysterol biology. However, several important revisions are required before the manuscript can meet academic publishing standards.

The abstract and introduction appear to be generally well-prepared. However, I strongly recommend including an explicit statement of the research aim in both sections (e.g., “The aim of the present study was to…”). A clearly formulated objective would significantly enhance the manuscript’s clarity and help readers better understand the authors’ intentions. Moreover, the final part of the introduction should be revised. It is not appropriate to include a summary of the study’s results in the introduction—this is not the abstract. This structure should be corrected.

The results are described in detail, but several corrections are necessary.

Please remove the redundant figure labels (e.g., “Figure 1,” “Figure 2”) from the top-left corners of the images; the figure numbers in the legends are sufficient.

Figure 1B: The line graph should be replaced with a bar chart, as the data represent discrete conditions rather than a continuous parameter.

Figure 2: The image lacks magnification information and a proper scale bar. Please refer to standard guidelines for determining magnification, such as those provided here: https://carson.com/optics-university/microscope-hub/how-to-calculate-the-magnification-of-a-microscope

Additionally, it is apparent that panels A and B in Figure 2 were taken at different magnifications and most likely using different microscopy systems. This should be clearly indicated in the figure legend or methods section.

Figure 4: Histogram data should be presented as a clearly labeled graph (e.g., panels A and B). Alternatively, retain only the graph in the main figure and move the original histograms to the Supplementary Information to improve readability.

Figure 6: It is unclear what data are shown in panels A and B. These panels should be more clearly labeled and described in the legend. As currently presented, it is difficult to interpret them correctly.

The methodology section is generally appropriate but requires further detail:

Please provide the final working concentrations of all reagents and staining dyes.

Dilutions and catalog numbers of primary and secondary antibodies should be specified.

Catalog numbers for all commercial assay kits used in the study must also be included.

Solvent controls (ethanol, DMSO) are not described. Were vehicle-only controls used? If not, this is a significant omission and should be corrected.

The manuscript would benefit from a graphical abstract or pathway schematic (e.g., new Figure 7) summarizing the mechanism: 25OHChol → mitochondria → Bax/Bcl-2 shift → caspase cascade → apoptosis.

The discussion and conclusion are balanced and well-structured. However, the manuscript has a major limitation: the mechanistic depth is insufficient. Although the involvement of caspases was demonstrated, the study does not go beyond this to explore more detailed molecular pathways. The lack of mechanistic insight significantly limits the scientific impact of the findings.

Author Response

< Answer for Reviewer 2 >

Review of the manuscript: “25-Hydroxycholesterol Induces Intrinsic Apoptosis via Mitochondrial Pathway in BE(2)-C Human Neuroblastoma Cells”. The submitted manuscript presents a carefully designed experimental study evaluating the cytotoxic effects of 25-hydroxycholesterol (25OHChol) on BE(2)-C neuroblastoma cells. The methodologies used—CCK-8, DAPI, Annexin V/PI, JC-1 staining, caspase activity assays, and Western blotting—are appropriate and convincingly support the activation of the intrinsic apoptotic pathway. The article is scientifically sound, well-documented, and potentially valuable to the fields of neuro-oncology and oxysterol biology. However, several important revisions are required before the manuscript can meet academic publishing standards.

The abstract and introduction appear to be generally well-prepared. However, I strongly recommend including an explicit statement of the research aim in both sections (e.g., “The aim of the present study was to…”). A clearly formulated objective would significantly enhance the manuscript’s clarity and help readers better understand the authors’ intentions. Moreover, the final part of the introduction should be revised. It is not appropriate to include a summary of the study’s results in the introduction—this is not the abstract. This structure should be corrected.

A: We fully agree with your suggestion. We have made the necessary revisions accordingly, and the modified sections are highlighted in blue. Thank you very much for your valuable and constructive feedback.

The results are described in detail, but several corrections are necessary.

Please remove the redundant figure labels (e.g., “Figure 1,” “Figure 2”) from the top-left corners of the images; the figure numbers in the legends are sufficient.

A: We agree with your advice. The points are corrected.

Figure 1B: The line graph should be replaced with a bar chart, as the data represent discrete conditions rather than a continuous parameter.

A: Yes, we corrected the points.

Figure 2: The image lacks magnification information and a proper scale bar. Please refer to standard guidelines for determining magnification, such as those provided here: https://carson.com/optics-university/microscope-hub/how-to-calculate-the-magnification-of-a-microscope.

Additionally, it is apparent that panels A and B in Figure 2 were taken at different magnifications and most likely using different microscopy systems. This should be clearly indicated in the figure legend or methods section.

A: Yes, we have made the necessary revisions and included the additions you suggested, and the modified sections are highlighted in blue.

Figure 4: Histogram data should be presented as a clearly labeled graph (e.g., panels A and B). Alternatively, retain only the graph in the main figure and move the original histograms to the Supplementary Information to improve readability.

A: In accordance with the reviewer’s suggestion, we have separated Figure 4A to improve its clarity. Likewise, we have revised Figure 6C in the same manner to enhance its clarity. Thank you.

Figure 6: It is unclear what data are shown in panels A and B. These panels should be more clearly labeled and described in the legend. As currently presented, it is difficult to interpret them correctly.

A: Yes, we added information of the panels.

The methodology section is generally appropriate but requires further detail:

Please provide the final working concentrations of all reagents and staining dyes.

A: Yes, we agree with the reviewer’s advice. The treatment concentrations of all reagents used have been specified in both the Materials and Methods section and the figure legends. Thank you.

Dilutions and catalog numbers of primary and secondary antibodies should be specified. Catalog numbers for all commercial assay kits used in the study must also be included.

A: Yes, we have incorporated the point you raised into the Materials and Methods section and highlighted it in blue.

Solvent controls (ethanol, DMSO) are not described. Were vehicle-only controls used? If not, this is a significant omission and should be corrected.

A: This is a very important point. Over the past decade, we have submitted numerous papers on oxysterols. In each case, we have consistently used ethanol—the solvent for cholesterol and oxysterols—as a control, and, as always, at the concentrations we apply, the results have shown no difference compared with the general control group. In this study, all oxysterols, including cholesterol, were dissolved in ethanol prior to treatment. However, as shown in Result 1A, apoptosis was observed only in the 25OHChol-treated group. This further reinforced our confidence that ethanol itself did not influence our results. Such results have routinely been included as supplementary data.

In the present manuscript, however, this process was inadvertently omitted, and we acknowledge that this was our oversight. At present, the research team that conducted the experiments for this study has been disbanded, and the review period is also quite tight. We kindly ask for your understanding in this regard.

In future experiments, we will take the reviewer’s valuable comment to heart and implement a double-checking process to ensure such an omission does not occur again.

Thank you.

The manuscript would benefit from a graphical abstract or pathway schematic (e.g., new Figure 7) summarizing the mechanism: 25OHChol → mitochondria → Bax/Bcl-2 shift → caspase cascade → apoptosis.

A: Thank you for the excellent suggestion. We will include it as a graphical abstract.

The discussion and conclusion are balanced and well-structured. However, the manuscript has a major limitation: the mechanistic depth is insufficient. Although the involvement of caspases was demonstrated, the study does not go beyond this to explore more detailed molecular pathways. The lack of mechanistic insight significantly limits the scientific impact of the findings.

A: Your insightful and precise comments have significantly contributed to elevating the quality of our manuscript. Although this study still has many areas that require improvement, we will keep your feedback firmly in mind to guide it toward a stronger and more refined work. We will also ensure that our subsequent research is reviewed and conducted with even greater care. We are truly grateful for the thoughtful and heartfelt advice you have provided.

Reviewer 3 Report

Comments and Suggestions for Authors

The manuscript entitled "25-Hydroxycholesterol Induces Intrinsic Apoptosis via Mitochondrial Pathway in BE(2)-C Human Neuroblastoma Cells " is an interesting attempt to gain more information on the cytotoxic effects of 24sOHChol, 25OHChol, and 27OHChol on BE(2)-C neuroblastoma cells and to elucidate the underlying mechanisms of their apoptotic activity.

The authors demonstrates that 25OHChol triggers apoptosis in BE(2)-C neuroblastoma cells through activation of the intrinsic mitochondrial apoptotic pathway.

The overall composition of the manuscript is good, with a very nice introduction to the topic. The quality of the figures and tables is also good. The paper is written in a clear and understanding manner and the results are properly presented.

Overall I have find this manuscript suitable for publication in “International Journal of Molecular Sciences” in current form.

Author Response

< Answer for Reviewer 3 >

The manuscript entitled "25-Hydroxycholesterol Induces Intrinsic Apoptosis via Mitochondrial Pathway in BE(2)-C Human Neuroblastoma Cells " is an interesting attempt to gain more information on the cytotoxic effects of 24sOHChol, 25OHChol, and 27OHChol on BE(2)-C neuroblastoma cells and to elucidate the underlying mechanisms of their apoptotic activity.

The authors demonstrates that 25OHChol triggers apoptosis in BE(2)-C neuroblastoma cells through activation of the intrinsic mitochondrial apoptotic pathway.

The overall composition of the manuscript is good, with a very nice introduction to the topic. The quality of the figures and tables is also good. The paper is written in a clear and understanding manner and the results are properly presented.

Overall I have find this manuscript suitable for publication in “International Journal of Molecular Sciences” in current form.

A: I sincerely appreciate your positive and encouraging review. I will strive to reciprocate with research of even higher quality. Thank you once again.

Round 2

Reviewer 2 Report

Comments and Suggestions for Authors

Authors revised manuscript according to suggestions